# Human vtRNA1-1 Levels Modulate Signaling Pathways and Regulate Apoptosis in Human Cancer Cells

**DOI:** 10.3390/biom10040614

**Published:** 2020-04-16

**Authors:** Lisamaria Bracher, Iolanda Ferro, Carlos Pulido-Quetglas, Marc-David Ruepp, Rory Johnson, Norbert Polacek

**Affiliations:** 1Department of Chemistry and Biochemistry, University of Bern, 3012 Bern, Switzerland; lisamaria.bracher@dcb.unibe.ch (L.B.); iolanda.ferro@dcb.unibe.ch (I.F.); marc-david.ruepp@kcl.ac.uk (M.-D.R.); 2Graduate School for Cellular and Biomedical Sciences, University of Bern, 3012 Bern, Switzerland; carlos.pulido@dbmr.unibe.ch; 3Department of Medical Oncology, Inselspital, University Hospital and University of Bern, 3010 Bern, Switzerland; rory.johnson@dbmr.unibe.ch; 4Department of BioMedical Research, University of Bern, 3008 Bern, Switzerland; 5United Kingdom Dementia Research Institute, King’s College London, Institute of Psychiatry, Psychology and Neuroscience, Maurice Wohl Clinical Neuroscience Institute, London SE5 9NU, UK

**Keywords:** non-coding RNA, vault RNA, apoptosis, signaling

## Abstract

Regulatory non-protein coding RNAs perform a remarkable variety of complex biological functions. Previously, we demonstrated a role of the human non-coding vault RNA1-1 (vtRNA1-1) in inhibiting intrinsic and extrinsic apoptosis in several cancer cell lines. Yet on the molecular level, the function of the vtRNA1-1 is still not fully clear. Here, we created HeLa knock-out cell lines revealing that prolonged starvation triggers elevated levels of apoptosis in the absence of vtRNA1-1 but not in vtRNA1-3 knock-out cells. Next-generation deep sequencing of the mRNome identified the PI3K/Akt pathway and the ERK1/2 MAPK cascade, two prominent signaling axes, to be misregulated in the absence of vtRNA1-1 during starvation-mediated cell death conditions. Expression of vtRNA1-1 mutants identified a short stretch of 24 nucleotides of the vtRNA1-1 central domain as being essential for successful maintenance of apoptosis resistance. This study describes a cell signaling-dependent contribution of the human vtRNA1-1 to starvation-induced programmed cell death.

## 1. Introduction

In multicellular organisms the ratio of non-coding to protein coding DNA drastically increases as compared to bacteria and archaea [1]. In fact, there are more genes encoding non-coding RNAs (ncRNAs) than genes coding for proteins in complex eukaryotes, including *Homo sapiens*, giving rise to a vast network of regulatory RNAs [2]. With the ever-continuing progress in deep sequencing techniques, the existence of thousands of ncRNAs candidates has been uncovered. For a fraction of these ncRNA candidates, biological functions in gene regulation have been identified at various levels, including transcription, RNA processing, genome stability, DNA replication, and translation [3]. However, for many potential ncRNAs, experimental validation is still pending.

Vault RNAs (vtRNAs) are ncRNAs initially identified in the mid 80s by the Rome lab as integral components of the large 13 MDa vault complex isolated from rat liver tissue [4], a highly conserved ribonucleoprotein complex (RNP) found in the cytoplasm of many eukaryotic species [5]. Several functional roles have been suggested for the vault complex including involvement in nucleo-cytoplasmic transport [6], the development of multi-drug-resistance (MDR) in cancer cells [7,8], cellular signaling [9,10], DNA damage repair [11], the innate immune response [12], apoptosis resistance [13], and nuclear-pore complex formation [14]. In humans, four expressed vtRNA paralogs have been detected, vtRNA1-1, vtRNA1-2, vtRNA1-3, and vtRNA2-1, ranging from 88 to 102 nucleotides in length [15,16]. The genes locate to chromosome 5 and are transcribed through polymerase III (polIII). Besides two internal polIII promoter elements, the A-box and B-box, only little sequence conservation is observed [16]. Strikingly, only about 5% of the total cellular vtRNA associates with the vault complex, whereas ≈ 95% is evenly distributed throughout the cytoplasm [7,16]. This raises the question about the functional relevance of this large fraction of vtRNA for the cell. To date, several functions of the vault complex-independent vtRNA have been suggested. These include direct binding of vtRNA1-1 and vtRNA1-2 to the chemotherapeutic compound mitoxantrone [17], its involvement in mitoxantrone resistance in glioblastoma, leukemia and osteocarcinoma derived cell lines [18], and micro RNA-like (miRNA) regulation of gene expression involved in MDR of breast cancer cells [19]. Loss of NSun2-mediated methylation of vtRNA1-1 C69 causes aberrant processing into small vtRNA fragments, shaping the epidermal differentiation program of keratinocytes in an argonaut-dependent manner. This process is counteracted by direct interaction of the unmethylated vtRNA1-1 with the serine/arginine rich splicing factor 2 (SRSF2) [20,21]. vtRNA1-1 is also involved in malignant cell proliferation of human breast cancer through direct interaction with the polypyrimidine tract binding splicing factor (PSF), thereby releasing oncogenic transcription repression [22]. vtRNA2-1, also known as nc886, was reported to physically interact with and repress the interferon-inducible protein kinase RNA-activated (PKR) and thereby preventing aberrant cell proliferation in lung epithelial cell lines [23]. In cholangiocarcinoma (CCA) nc886 is repressed causing PKR activation and subsequent pro-survival NF-κB activation, whereas vtRNA2-1 alone is sufficient to repress PKR activation in non-malignant cholangiocytes, suggesting a role for vtRNA2-1 in tumor surveillance and elimination of pre-malignant cell during tumorigenesis [24]. Recently, the existence of vtRNA in the parasitic kinetoplastid *Trypanosoma brucei* was confirmed. In association with *T. brucei* TEP1 it occupies a central role in trans-splicing of the parasite’s mRNAs [25,26]. In human hepatocytes vtRNA1-1 binding to the central autophagy receptor SQSTM1/p62 was reported to regulate autophagy by preventing SQSTM1/p62 self-oligomerization [27]. Previously, we demonstrated that expression of vtRNA1-1, but not of the other vtRNA paralogs, resulted in increased apoptosis resistance in a model cell line of Burkitt Lymphoma [28]. This anti-apoptotic effect was shown to be unaffected by major vault protein (MVP) levels, with MVP contributing with 70% to the mass of the vault complex, thus hinting at a vault particle-independent regulatory function of vtRNA1-1. vtRNA1-1 mediated apoptosis resistance was evident also in HeLa and breast cancer cells [28]. Yet, the molecular function of the vtRNA1-1 in apoptosis resistance is still not fully clear.

Cells receive and respond to extracellular stimuli usually through receptors that, once triggered, integrate the extracellular signals into a nexus of fine-tuned intracellular signaling networks, resulting ultimately in a transcriptional response shaping the fate of the cell population [29]. The extracellular receptors are categorized into G-protein coupled receptors (GPCRs), receptor tyrosine kinases (RTKs), and ionotropic receptors. The cytoplasmic receptor domains serve as the anchor point for signal transducing enzymatic complexes consisting of distinct kinases, phosphatases, and other molecules and are responsible for the downstream signal transduction, often through the production of second messenger molecules including phosphatidylinositol-3,4,5-triphosphate (PIP_3_) or cyclic AMP (cAMP). Typical signal transducing proteins are heteromeric G-proteins that may initiate cAMP downstream of GPCRs or the phosphatidylinositol-3-kinase (PI3K) isoforms catalyzing the phosphorylation of phosphatidylinositol-4,5-bisphosphate (PIP_2_) into PIP_3_ [30]. Signals can be transduced via several intermediary transducing proteins, yet the final targets of every signaling network are transcription factors, which convert the initially encountered stimulus into an alteration of the cellular activity through determination of the final cellular gene expression. Cellular signaling pathways are frequently interacting with each other and post-translational modifications, like phosphorylation or methylation, expand the complexity [29].

In this study we generated individual HeLa vtRNA1-1 and vRNA1-3 knock-out cell lines and induced apoptotic cell death via serum deprivation and simultaneous inhibition of autophagy. This resulted in a strong decrease of apoptosis resistance specific to the loss of vtRNA1-1. A very similar dependency of apoptosis resistance on vtRNA1-1 levels was also revealed in two other immortalized human cell lines (HEK293 and A549). The total mRNA transcriptome of HeLa wild-type (WT), vtRNA1-1 knock-out, vtRNA1-3 knock-out, and vtRNA1-1 complementation cells was investigated upon apoptosis induction. Thereby, we identify the PI3K/Akt pathway and the ERK1/2 MAPK cascade to be misregulated upon vtRNA1-1 loss. A mutational screen of vtRNA1-1 variants revealed a short stretch within the central domain as essential for conferring apoptosis resistance.

## 2. Materials and Methods

### 2.1. Cell Culture

The human embryonic kidney-derived cell line HEK293T and HeLa cells (ATCC^®^-CCL-2™), adherently growing cervical adenocarcinoma cells, were cultured in Dulbecco’s modified Eagle’s medium (DMEM/F12) supplemented with 10% fetal bovine serum, 292 µg/mL L-glutamine, and antibiotics (100 U/mL penicillin and 100 µg/mL streptomycin). After genome editing (see below) successfully edited HeLa cells were selected for 48 h with 1.5 µg/mL puromycin. Following lentiviral transduction (see below), HeLa cells were selected for at least 7 days with 5 µg/mL puromycin.

### 2.2. MTT Proliferation Assay

The assessment of metabolic activity was performed in 96-well plates. HeLa cells were seeded as triplicates with a density of 5000 cells/well and allowed to recover overnight. Cell proliferation was measured at different time points by replacing the growth medium with 100 µL growth medium supplemented with 10% sterile filtered MTT (stock solution: 5 mg/mL in PBS) and incubated at 37 °C for 4 h. The medium was replaced with 50 µL DMSO and subsequently incubated at 37 °C for 10 min. Absorbance was measured at 570 nm using a plate reader.

### 2.3. Genome Editing

Single guide RNAs (sgRNAs) targeting the up- and downstream regions of the vtRNA1-1 and the vtRNA1-3 loci (Appendix A) were predicted using the CRISPR online tool (crispr.mit.edu) from the Zhang lab [31]. Matching oligonucleotides including 5′ *BbsI* overhangs were ordered from Microsynth AG (Balgach, Switzerland) phosphorylated, annealed, and cloned into pCRISPR-EF1α-eSpCas9(1.1) using the *BbsI* restriction sites [32,33]. Individual sgRNA constructs were tested for eSpCas9 cleavage efficiency by transfection of 1 µg plasmid DNA into 2 × 10^5^ HeLa cells followed by 48 h eSpCas9 cleavage. Following genomic DNA (gDNA) extraction, vtRNA loci specific cleavage of gDNA was tested with the GeneArt genomic cleavage detection kit (Thermo Scientific: Waltham, MA, USA) following the manufacturer’s published protocol. Efficiency validated sgRNAs were PCR amplified (including the U6 promoter) using the primers U6-EcoRI-fwd/gRNA-XbaI-rev (Appendix A) and cloned into pCRISPR-EF1α-eSpCas9(1.1) upstream of the already present sgRNA insert, using the *EcoRI* and *XbaI* sites, thus creating plasmid constructs containing one locus upstream targeting sgRNA, one locus downstream targeting sgRNA and the eSpCas9 (Appendix A). The vtRNA1-1 and the vtRNA1-3 loci including the genomic sequence between the two targeted PAM sites were PCR amplified with primers HVG1-us2_SalI-fwd/HVG1-ds2_SpeI-rev, HVG1up1-SalI-fwd/HVG1dw1-SpeI-rev or HVG3up2-SalI-fwd/HVG3dw1-SpeI-rev (Appendix A) and cloned into the genomic cleavage selection plasmid pMB1610-pRR-puro [34] using the *SalI* and *SpeI* restriction sites (Appendix A). Then, 2.5 × 10^5^ HeLa cells were transiently co-transfected with 0.5, 1.0, or 2.0 µg of pCRISPR-EF1α-eSpCas9(1.1) and 200 ng of pMB1610-pRR-EF1α-puro with the JetPEI DNA transfection reagent (Polyplus transfection: New York, NY, USA). Genomic cleavage was performed for 48 h and subsequently the growth medium was replaced with medium supplemented with 1.5 µg/mL puromycin and cells selected for 48 h. Single cell colonies were attained by plating successfully selected cells in large 150 cm^2^ cell culture dishes, where they could form single cell colonies within 1–2 weeks. Individual colonies were picked and propagated until sufficient total RNA could be extracted (TriReagent protocol; Lucerna Chem AG: Luzern, Switzerland) for northern blot analysis (see below).

### 2.4. Cloning

vtRNA1-1 central domain mutants were generated from the Gateway cloning system entry plasmid pENTR-U243-vtRNA1-1 cloned previously [28], either with conventional PCR, QuickChange mutagenesis, or PIPE cloning. The listed primers (Appendix A) were used to generate mutants M1 to M8. After successful mutation, each entry plasmid was recombined in a Gateway LR-reaction with the lentiviral destination vector pHR-dest-SFFV-puro.

### 2.5. Lentiviral Transduction

Lentiviral particles were produced in human HEK293T cells, which were transiently transfected with lentiviral constructs containing cDNAs coding for all gene products mentioned in the text, together with the packaging plasmids pSPAX and the envelope plasmid pVSV-G. After 48 and 72 h, lentiviral supernatant was collected, sterile filtered (Whatman filter 0.2 µM, GE Healthcare: Chicago, IL, USA), and supplemented with polybrene to a final concentration of 4 µg/mL and added to the target cells overnight.

### 2.6. Annexin V Staining and Flow Cytometry

Following growth in starvation medium (DMEM/F12, 0.1% fetal bovine serum, 20 µM chloroquine (Sigma-Aldrich: St. Louis, MO, USA) for 24 or 48 h, cells were washed twice in ice-cold PBS, detached with trypsin for 2 min, collected by centrifugation (200 g, 5 min) and resuspended in 100 µL 1× Annexin binding buffer. Then, 2.5 µL Annexin V-PE and 2.5 µL 7-AAD were added to the cells, which were then incubated for 15 min in the dark. Next, 400 µL 1× Annexin binding buffer was added (PE-Annexin V Apoptosis Detection Kit I, BD Pharmingen: San Diego, CA, USA). The samples were analyzed by flow cytometry within 1 h on a LSRII SORP H271 or an LSRII SORP H274 FACS device and gated based on forward versus side scatter for size. The number of early apoptotic cells was assessed by recording the PE-YG channel (561 nm), and the number of Annexin V positive cells by recording the 7-AAD channel (488 nm). Cells double positive for Annexin V and 7-AAD were in late apoptosis. Data was evaluated using the FlowJo V10 software.

### 2.7. Northern Blot

Five to ten micrograms of total RNA (TriReagent protocol, Lucerna Chem AG: Luzern, Switzerland) isolated from HeLa cells and derivates thereof were separated on 8% or 10% denaturing polyacrylamide gels (7 M urea, 1× TBE buffer), transferred onto nylon membranes, UV-crosslinked and probed with 5′-^32^P-end labeled antisense DNA probes (Appendix A) as described [35]. Northern blot signals were visualized on a phosphoscreen and quantified using the ImageQuantTL software. All quantified northern blot signals were normalized to the band intensities of the 5.8S rRNA.

### 2.8. Western Blot

Cells were lysed in 1× RIPA buffer (50 mM Tris-HCl pH 8.0, 150 mM NaCl, 1% Igepal CA-630, 0.5% sodium deoxycholate and 0.1% SDS) and protein extracts supplemented with 6× SDS loading dye (375 mM Tris-HCl pH 6.8, 6% SDS, 30% glycerol, 30% β-mercaptoethanol, and 0.03% bromophenol blue). Equal amounts of protein (30–50 µg, determined with the BCA Protein assay kit, Thermo Scientific: Waltham, MA, USA) were separated on 10% denaturing SDS polyacrylamide gels and blotted onto pure nitrocellulose membranes (unsupported, 0.45 µM, AppliChem GmBH: Darmstadt, Germany). Membranes were blocked for 1 h at room temperature in 1× TBS with 0.1% Tween 20 and 5% nonfat dry milk and incubated with the primary antibodies overnight at 4 °C. Horseradish peroxidase conjugated secondary antibodies were added for 1 h, IRDye conjugated secondary antibodies (Li-Cor) were added for 45 min, both at room temperature. Immune complexes were visualized using the SuperSignal West Femto maximum sensitivity substrate (Thermo Scientific: Waltham, MA, USA) or were visualized with the Odyssey imaging system according to the manufacturer’s instructions. The following primary antibodies and dilutions were used: rabbit anti-Gapdh 1:3000 (14C10, Cell Signaling: Danvers, MA, USA); mouse anti-p44/42 1:2000 (L34F12, Cell Signaling); phospho-ERK1/2 pathway sampler kit (#9911, Cell Signaling); phospho-Akt pathway sampler kit (#9916, Cell Signaling).

### 2.9. Next Generation mRNA Sequencing

Triplicates of HeLa WT, HeLa vtRNA1-1 knock-out, HeLa vtRNA1-3 knock-out, and HeLa vtRNA1-1 complementation cells were grown under starvation conditions or control conditions for 24 h. Efficiency of the treatment was confirmed by Annexin V staining and flow cytometry. Total RNA was extracted (TriReagent protocol, Lucerna Chem AG: Luzern, Switzerland) and additionally phenol-chloroform-isoamyl alcohol purified. Then, 5 µg of RNA was DNAse I (Thermo Scientific: Waltham, MA, USA) digested. Samples were then directly submitted to the next generation sequencing platform of the University of Bern, where poly(A) RNAs were enriched and RNA samples conversed into a library of template molecules with known strand origin (TruSeq stranded mRNA sample preparation kit, Illumina). For the 150 base pair paired-end sequencing using an Illumina HiSeq3000 device, the libraries were additionally size selected. mRNA-Seq sequencing data have been deposited at GEO database under the accession number GSE147054.

### 2.10. Bioinformatic Analysis

The quantification of the different transcript abundances was computed using the Kallisto software v0.44.0 [36] (-b 100 --rf-stranded) based on transcripts of the GENCODE annotation (GRCh38.p12). Kallisto performs a pseudo-alignment using the raw RNA-Seq data, therefore no mapping step is required prior to the quantification. The average number of paired-end reads per sample was 92.5 ± 25.1 million. Reads demonstrated roughly 50% GC content and lengths of 140–151 nucleotides. The average of pseudo aligned reads by Kallisto was 82 ± 23 million (roughly 90% of total reads per sample) from which an average of 23 ± 7 million were unique mapped reads. Transcripts Per kilobase Million (TPM) values for each transcript were used to compute differential gene expression (DE) with the Sleuth software v 0.30.0 [37]. A minimum of five reads per transcript in at least one treatment group was required to include the transcript in the DE analysis. Genes with Log2FC ≥ 1 and *p* values ≤ 0.01 were classified as differentially expressed.

## 3. Results

### 3.1. vtRNA1-1 Knock-Out Stimulates Starvation-Induced Apoptosis

Previously, we have shown that ectopic expression of human vtRNA1-1 but not its paralogs vtRNA1-2 or vtRNA1-3 in Burkitt Lymphoma derived BL41 cells results in significantly increased apoptosis resistance to intrinsic and extrinsic stimuli [28]. Furthermore, the reduction of endogenous vtRNA1-1 using an anti-sense oligonucleotide (ASO) approach leads to more spontaneous cell death in HeLa cells and the breast cancer cell line HS578T. Thus, indicating an involvement of vtRNA1-1 in apoptosis resistance of different cancer cell types. A similar effect was observed upon ASO-mediated vtRNA1-1 depletion in HEK293 cells and in the lung cancer cell line A549 (Appendix A). In the A549 cells the loss of apoptosis resistance was further stimulated when the cells were challenged with staurosporine, a well-established pan-kinase inhibitor and inducer of intrinsic apoptosis (Appendix A). These experiments substantiate the previous findings and indicate a rather general function of the vtRNA1-1 in programmed cell death. In order to shed more light on the molecular function of vtRNA1-1 in a clean genetic background, CRISPR/Cas9 based genome editing was applied in HeLa cells. We successfully generated individual vtRNA1-1 and vtRNA1-3 knock-out cells by targeting both vtRNA genes individually with two single guide RNAs (sgRNAs) at a time, resulting in a complete clearance of the transcriptional units from the HeLa cell genome (Figure 1a,b; Appendix A).

By using stable lentiviral transductions, we successfully complemented the vtRNA1-1 expression, albeit at a lower level, in the HeLa vtRNA1-1 knock-out cells (Figure 1b; Appendix A). Subsequently to genome editing, the new cell lines were characterized based on cell morphology, proliferation rate, and migratory behavior to ensure healthy growth under standard conditions. Indeed, the vtRNA1-1 KO cells did not show striking morphological differences when compared to HeLa WT cells (Figure 1c). Cell proliferation was continuously monitored up to 72 h using an MTT assay. This analysis revealed a minor growth deficit of all cell lines subjected to CRISPR genome editing compared to the untreated parental HeLa WT control, suggesting a genome editing-dependent effect (Figure 1d; Appendix A). However, migration assays revealed no noticeable discrepancies of their invasive behavior between the HeLa vtRNA1-1 knock-out and the HeLa WT cells over the course of 24 h (Appendix A). To validate the effect of the vtRNA1-1 knock-out on general apoptosis resistance, we challenged the cells with growth serum deprivation combined with inhibition of the autophagic flux by chloroquine. For that, cells were grown for 24 or 48 h in starvation medium and subsequently subjected to the Annexin V binding assay and flow cytometric quantification. The loss of vtRNA1-1 but not of vtRNA1-3 expression resulted in an apparent increase of apoptotic cells after 24 h (Appendix A) and was further enhanced after 48 h of treatment (Figure 1e; Appendix A). Reintroduction of vtRNA-1-1 expression in the vtRNA1-1 complementation cell line could rescue the apoptosis rates to almost wild-type levels (Figure 1e; Appendix A). In summary, these data strengthen the causal link between apoptosis resistance and vtRNA1-1 expression and strongly indicate a specific role of vtRNA1-1 in starvation-induced programmed cell death, which was demonstrated in a clean genetic background completely devoid of endogenous vtRNA1-1.

### 3.2. mRNA Deep Sequencing Reveals a vtRNA1-1 Dependent Increase in Gene Expression of Cellular Signaling Activity

To investigate the role of vtRNA1-1 in starvation-induced apoptosis on a global scale, changes in the mRNA transcriptome levels were assessed with Illumina based next generation sequencing. HeLa WT, vtRNA1-1 knock-out, vtRNA1-1 complementation cells as well as vtRNA1-3 knock-out cells were cultivated under starvation conditions for 24 h. At this time point vtRNA1-1 knock-out cells demonstrate programmed cell death rates of ≈45% Annexin V positive cells (Appendix A). Subsequently, total RNA was isolated and submitted for poly(A) enrichment, cDNA library preparation, and Illumina deep sequencing. A T-distributed Stochastic Neighbor Embedding (t-SNE) analysis of the mRNAseq data performed with all genes confirmed the quality and similarity of the corresponding sample replicas (Figure 2a). vtRNA1-1 knock-out control sample #15 and vtRNA1-1 knock-out starvation sample #17 were removed for further downstream analyses because they obviously do not cluster with the corresponding replicas (Figure 2a). HeLa WT and vtRNA1-1 complementation samples exhibited a very similar clustering, confirming a WT-like gene expression pattern in the complementation cell line (Figure 2a). Unexpectedly, the vtRNA1-1 knock-out and the vtRNA1-3 knock-out also demonstrated a very similar clustering between each other, although the pattern was clearly distinct from the control samples (Figure 2a). The sequencing data was analyzed with a differential expression (DE) analysis comparing the individual cell lines. By taking only transcripts into consideration that showed a *p*-value ≤ 0.01 and a fold-change of at least 2-fold, data complexity was reduced to 623 DE genes between HeLa WT and vtRNA1-1 knock-out cells and 634 DE genes between HeLa WT and vtRNA1-3 knock-out cells following 24 h of starvation, respectively (Appendix A Dataset S1).

Further comparison of these two data sets resulted in 205 upregulated DE transcripts specific for the vtRNA1-1 knock-out under starvation from which 167 genes could be annotated in the database for annotation, visualization, and integrated discovery (DAVID). Gene ontology (GO) analysis revealed that many of the enriched genes encode proteins involved in cellular signaling (Figure 2b; Appendix A). Comparing control and starvation conditions in the HeLa vtRNA1-1 knock-out cells, several genes annotated to two prominent signaling cascades involved in cell survival and proliferation, the PI3K/Akt pathway and the MAPK cascade, are differentially expressed (Figure 3a,c). When comparing control and starvation conditions within one cell line (Figure 3a,c), the volcano plots only reveal significantly differentially expressed genes as a consequence of starvation. For the PI3K/Akt pathway, ITGA2, ITGA10, LPAR1, and LAMC2 are markedly upregulated upon starvation (Figure 3a), whereas for the MAPK cascade the gene expression of DUSP-5, DUSP-6, EREG, and RASGRF2 is significantly stimulated (Figure 3c). However, for some genes annotated to either the PI3K/Akt pathway or the MAPK cascade, the loss of vtRNA1-1 expression was sufficient to induce increased expression which the starvation treatment could only mildly stimulate further (Figure 3b,d). This is the case for the genes EPHA2, PGF, CSF1, or PAK3, annotated to the PI3K/Akt or the MAPK signaling, respectively (Figure 3b,d). This finding suggests that the expression of these genes is specific for the loss of vtRNA1-1 but not necessarily specific for starvation (Figure 3b,d). To gain further insight into the observed increased signaling activity in the HeLa vtRNA1-1 knock-out cells under starvation, we decided to look in more detail at the PI3K/Akt pathway and the phospho-ERK1/2 MAPK cascade.

### 3.3. The Loss of vtRNA1-1 Stimulates PI3K/Akt Pathway Signaling

To confirm the mRNAseq results we decided to investigate the stimulation of the PI3K/Akt signaling in vtRNA1-1 knock-out conditions when cells are suffering from growth factor withdrawal followed by recovery in complete growth serum. Doing so, HeLa WT, vtRNA1-1 knock-out, and vtRNA1-1 complementation cells were grown in starvation medium for 24 h, then the medium was replaced with complete 10% FBS containing medium and cells were allowed to recover from starvation for different periods of time. Whole cell protein extracts were subjected to western blot analysis, assessing the phosphorylation state of major members of the PI3K/Akt pathway. The pathways main kinase Akt is induced by a phosphorylation of Thr308 of its hydrophobic motif through the PIP_3_ secondary messenger activated kinase PDK1 [38] (Figure 4a). A second phosphorylation in the hydrophobic motif by mTORC2 at Ser473 is necessary for maximal Akt activation [39] (Figure 4a). The plasma membrane lipid phosphatase tumor suppressor phosphatase and tensin homolog deleted on chromosome 10 (PTEN) is the major antagonist of the PI3K/Akt signaling when unphosphorylated [40]. In untreated control conditions and after 24 h of starvation, mTORC2 mediated Akt phosphorylation at Ser473 is not detectable in any of the three tested cell lines (Figure 4b). However, following 20 min of pathway induction by serum re-administering, the vtRNA1-1 knock-out cells demonstrate a much stronger pAkt Ser473 activation compared to the HeLa WT and the vtRNA1-1 complementation cells (Figure 4b), which ceases during the time of recovery until the phosphorylation levels almost completely disappeared after 24 h. Exactly the opposite effect is observed for the HeLa WT cells, revealing the highest Ser473 phosphorylation after 24 h of serum re-feeding (Figure 4b). Akt phosphorylation at Thr308 was not further followed in this context due to the absence of starvation-dependent changes under the applied conditions. Interestingly, the lack of vtRNA1-1 results in an enhanced PTEN Ser380 phosphorylation when cells were untreated (Figure 4b). While the amount of Ser380 phosphorylation correlates with the activation of Akt in the HeLa WT cells during the re-feeding phase, phospho-PTEN decreases during Akt Ser473 phosphorylation when vtRNA1-1 is lacking. In addition, once Akt Ser473 phosphorylation decreases, phospho-PTEN quantities increase again (Figure 4b). The vtRNA1-1 complementation cells demonstrate a comparable PTEN phosphorylation to vtRNA1-1 knock-out cells when unstimulated and the phosphorylation also decreases during the recovery phase, but after 24 h of normal growth phospho-PTEN levels are not re-established (Figure 4b). In order to induce the PI3K/Akt pathway more directly, Hela WT, vtRNA1-1 knock-out, and vtRNA1-1 complementation cells were starved for 20 min. The lack of growth factors reduced the levels of mTORC2-mediated Akt Ser473 phosphorylation drastically in all three cell lines, but least in the vtRNA1-1 knock-out cells (Figure 4c). In accordance to the re-feeding experiment (Figure 4b), PTEN displayed the same hyper-phosphorylation trend upon pathway stimulation (Figure 4c). Basal phospho-PTEN levels were clearly enhanced in the vtRNA1-1 knock-out cells compared to the Hela WT cells. While the phosphorylation levels in the HeLa WT do not change largely after only 20 min of starvation, they were clearly increased when vtRNA1-1 is not present (Figure 4c). Upon complementation of the vtRNA1-1 expression, PTEN Ser380 phosphorylation decreased when the pathway was induced, and the levels were more similar to those in HeLa WT cells. Taking these findings together, loss of vtRNA1-1 expression appears to mildly stimulate the basal activity of the pro-survival PI3K/AKT pathway and this effect is enhanced under detrimental growth factor deprivation.

### 3.4. The ERK1/2 Cascade Initiation in Starvation-Induced HeLa Cells Is Increased upon vtRNA1-1 Knock-Out

We next addressed the question whether the expression of vtRNA1-1 has also an impact on the extracellular signal regulated kinase cascades. For that, we investigated the phosphorylation state of the ERK1/2 pathway in HeLa WT, vtRNA1-1 knock-out, and vtRNA1-1 complementation cells as a consequence of pERK1/2 cascade induction. After receptor stimulation, c-Raf is activated through small membrane associated GTPases. c-Raf, in turn, triggers the kinases MEK1/2 that subsequently activate the pathway’s principal kinases ERK1/2 and hence the regulation of many transcription factors either directly through ERK1/2 or by one of its downstream targets MSK1 or p90RSK [41] (Figure 5a).

HeLa WT, vtRNA1-1 knock-out, and complementation cells were starved and allowed to recover analogously as described in the preceding paragraph. The loss of vtRNA1-1 results in an apparent stimulation of the activating c-Raf phosphorylation at Ser338 under control and starvation conditions when compared to HeLa WT and vtRNA1-1 complementation cells (Figure 5b). Ser338 phosphorylation was increased in all three cell lines 20 min after addition of the normal growth medium (Figure 5b) although only the vtRNA1-1 knock-out cells displayed a prolonged c-Raf stimulation during recovery (Figure 5b). Activation of ERK1/2 by phosphorylation of Thr202/Tyr204 was at comparable low levels when cells were untreated or starved for 24 h (Figure 5b). Upon pathway stimulation phospho-ERK1/2 amounts were increased in all cells but least in the vtRNA1-1 knock-out cell line (Figure 5b). During the recovery phase, cells lacking vtRNA1-1 expression displayed a less prolonged ERK1/2 stimulation compared to the vtRNA1-1 expressing cells (Figure 5b). When the ERK1/2 MAPK cascade was directly stimulated by cell starvation for 20 min and samples analyzed by western blot for phospho-protein levels of pathway involved proteins, again loss of vtRNA1-1 stimulated basal phospho-c-Raf Ser338 levels (Figure 5c) and were constant after 20 min of growth factor withdrawal (Figure 5c). In contrast to the re-feeding experiment (Figure 5b), phospho-ERK1/2 fraction was clearly larger in the cells devoid of vtRNA1-1 under starvation when compared to the HeLa WT and vtRNA1-1 complementation cells (Figure 5c). Concomitantly, total ERK1/2 protein expression remained remarkably constant in all samples (Figure 5c, pan-ERK1/2). In conclusion the western blot results are in line with the trend apparent in the transcriptome analysis about an activated ERK1/2 MAPK pathway in the absence of vtRNA1-1.

### 3.5. Only 24 Nucleotides within the vtRNA1-1 Central Domain Are Sufficient to Maintain Apoptosis Resistance

The data shown above suggest a role for vtRNA1-1 in modulating signaling pathways eventually orchestrating apoptosis. In order to unravel the essential sequence motif or secondary structure elements of the vtRNA1-1 in apoptosis resistance, we performed a mutagenesis approach. The human vault RNA paralogs are highly similar in secondary structure and differ mainly in the central domain [15]. We previously demonstrated by domain swapping that this central domain of vtRNA1-1 is responsible for preventing cells from undergoing apoptosis [28]. To identify the exact sequence within the central domain (nucleotides U21–C75) contributing to the anti-apoptotic phenotype, vtRNA1-1 central domain mutants were generated and stably expressed in HeLa vtRNA1-1 knock-out cells (Figure 6a,b). That way, the effect of the ectopically expressed mutant constructs on apoptosis can be investigated without interference from endogenous wild type vtRNA levels. The nucleotide sequence of the “central bulge” (A44–A49) was changed by purine-pyrimidine substitutions (M1). The mfold software predicted stem-loop G50–U67 was truncated by three base pairs and the single stranded loop sequence was mutated (M2). Two sub-mutants of M2 were created, one mutant that only shortens the stem loop (M3), and the second mutant encoding a purine-pyrimidine substituted single stranded loop (M4). To further test the importance of the stem-loop secondary structure M5 was generated, disrupting the stem. As a control, the stem was reconstituted by compensatory mutations (M6). In order to ensure accurate cellular expression, M2–M6 do not distort the internal promoter sequence of the B-box motif facilitating precise polymerase III transcription. The mfold software predicted stem loop C30–G43 was locked in a single stranded open confirmation by purine-pyrimidine exchange of nucleotides G32, U33, G43, and A45 (M7). The stem was reconstituted by M8, simultaneously exchanging purines to pyrimidines from C30–A40.

All mutant constructs were expressed at satisfactory levels (Figure 6b). First, we validated comparable growth of all lentivirally transduced cells with the MTT proliferation assay. In accordance to the vtRNA1-1 complementation cell line, mutant cells demonstrated a normal distribution of proliferation variability. Only M4 was growing significantly slower, however, this growth defect did not interfere with the following apoptosis assays (Figure 6c). Mutant vtRNA1-1 expressing HeLa cells were grown in starvation medium for 48 h and apoptotic cells were quantified by Annexin V staining followed by flow cytometry. This analysis revealed that only M1 and M2 were unable to compensate for the loss of wild-type vtRNA1-1 (Figure 6d,e). These results are suggesting that the sequence between nucleotides A44 and U67 is crucial for the maintenance of cell survival under starvation conditions. When only the stem was truncated or only the loop was mutated (M3, M4), the constructs were able to rescue the apoptosis resistance phenotype almost as effectively as the wild-type vtRNA1-1 transcript (Figure 6e). Rather to our surprise, the structural disruption of the important stem-loop (M5) behaved like wild-type vtRNA1-1 and cells displayed an even lower apoptosis death rate than the real wild-type vtRNA1-1. To examine whether the two vtRNA1-1 mutants M1 and M2 that are unable to rescue the HeLa vtRNA1-1 knock-out cells from undergoing apoptosis, show similar PI3K/Akt and ERK1/2 pathway activation, we tested the phosphorylation levels of representative pathway proteins by western blot analyses. Indeed, after 20 min of starvation, the phosphorylation pattern evident in the M1 and M2 mutants was basically identical to the vtRNA1-1 knock-out cell line and markedly deviated from the wild-type control (Appendix A). These findings together provide further evidence that the anti-apoptotic effect of vtRNA-1-1 is intrinsic to the central domain, more precisely to only 24 nucleotides within this domain. vtRNA1-1 derivatives carrying mutations in these 24 nucleotides, provoke a similar cellular signaling misregulation of the PI3K/Akt pathway and the ERK1/2 cascade as compared to cells completely devoid of vtRNA1-1 expression.

## 4. Discussion

Despite being identified already 35 years ago the biological and molecular functions of the vtRNA or the vault complex remain largely elusive (reviewed in [42]). More than 90% of the cellular vtRNA transcripts have been shown not to be associated with the vault complex [16] thus suggesting the existence of possible physiological roles for this class of ncRNAs beyond the vault complex. One of the most recent reports on vtRNA functions demonstrated an interaction of vtRNA1-1 with the autophagy receptor sequestosome-1/p62 in the human hepatocellular carcinoma cell line HuH-7 [27]. Upon p62 binding vtRNA1-1 was reported to regulate the p62-dependent autophagy. It was furthermore shown that decreasing vtRNA1-1 levels concomitantly lowered p62 levels as a consequence of increased autophagic flux [27]. In our study presented herein, we do not see a comparable co-dependence of vtRNA1-1 (Figure 1b) and p62 levels (Appendix A) in the cervical carcinoma HeLa cells. Therefore, it is possible that vtRNA possess cell-line specific roles and/or functions also beyond autophagy regulation. Indeed, we previously [32] and within this study showed a link between high vtRNA1-1 levels and apoptosis resistance in Burkitt Lymphoma cell lines (BL2, BL41), in a breast cancer model (HS578T), in human embryonic kidney cells (HEK293), in lung carcinoma cells (A549), and HeLa cells. The complete genomic deletion of vtRNA1-1 results in a strong decrease of cellular resistance to programmed cell death in HeLa cells (Figure 1e, Appendix A). Identical phenotypes were observed in independent knock-out and complementation clones thus excluding clonal peculiarities of the first set of cell lines due to putative CRISPR-triggered artefacts (Appendix A). These effects are specific since removal of vtRNA1-3 did not result in altered apoptosis rates. The anti-apoptotic effect of elevated vtRNA1-1 levels on programmed cell death is obvious when apoptosis is either triggered by well-known inducers of the intrinsic (staurosporine, etoposide) or extrinsic (Fas ligand) pathways [28], or by starvation (this study, Figure 1). By concomitant addition of the autophagy inhibitor chloroquine to the starvation medium allowed us to study a purely apoptotic phenotype.

Analysis of a global scale mRNA sequencing experiment performed in HeLa vtRNA1-1 and vtRNA1-3 knock-out cells demonstrated a differential expression (DE) of numerous genes involved in apoptosis, signal transduction, G-protein coupled receptor signaling, the MAPK cascade, the PI3K/Akt and the Rap1 pathway (Figure 3, Appendix A). During serum starvation with chloroquine the ERK1/2 MAPK cascade and the PI3K/Akt pathway are stimulated in the absence of vtRNA1-1 expression, shaping the cellular response to nutrient deprivation resulting in an increased apoptosis potential. However, the observation that these two signaling routes are more strongly activated when vtRNA1-1 is lacking was not necessarily anticipated, because the cellular default state of both signaling pathways is pro-survival. Thus, an enhanced activity of a survival pathway resulting in more cell death is counterintuitive at first.

Considering the mRNA sequencing data, most DE genes annotated to either the PI3K/Akt pathway or the MAPK cascade, locate functionally upstream of the central kinases Akt or ERK1/2, respectively. The DE genes are associated with signaling complexes close to or directly at the plasma membrane and the extracellular matrix organization in the case of PI3K/Akt signaling. Three genes specifically upregulated under starvation in the absence of vtRNA1-1 that annotate to the PI3K/Akt pathway are the cell surface receptors EPHA2, ITGA2, and ITGA10 (Figure 3a,b). In addition, our transcriptome analysis revealed the PI3K/Akt-specific upregulation of the laminin gene LAMC2 (Figure 3a,b). The loss of vtRNA1-1 resulted also in the upregulation of two soluble factors, CSF1 and PGF. The cytokine macrophage colony-stimulating factor 1 (CSF1) is known to induce PI3K/Akt signaling when binding its cognate RTK CSF1R in order to maintain survival of macrophages [43] whereas the placenta growth factor (PGF) is crucial for endothelial cell proliferation (Figure 3a,b). DE genes that were annotated to the activity of the MAPK cascade included the two dual specificity phosphatases, the nucleus located DUSP-5 and the cytoplasmic DUSP-6 (Figure 3c,d). Both regulate the MAPK activity on the level of ERK dephosphorylation and subsequent inactivation [44]. Furthermore, DE analysis revealed the serine/threonine protein kinase 3 (PAK3) that directly phosphorylates ERK4 and ERK6 following the activation by the small GTPases RAC1 and CDC42 (Figure 3c,d). The Ras-specific guanine nucleotide-releasing factor 2 (RASGRF2) is a Ca^2+^ dependent nucleotide exchange factor activating the two small GTPases RAC1 and Ras that in turn induce several pathways [45], including ERK1/2 signaling (Figure 3c,d). The ligand EREG binds members of the epidermal growth factor receptor family (ErbB receptors), thus stimulating overall intracellular signaling activity [46], including the MAPK (Figure 3c,d). Another pathway that appears to be upregulated upon vtRNA1-1 deletion is the small GTPase Rap1 signaling pathway (Figure 2b, Appendix A). Signaling via small GTPases like Ras or Rap1 usually relays extracellular signals from receptor tyrosine kinases into MAPK cascade signal transduction. We uncovered upregulation of the two nucleotide exchange factors RASGRF2 and RAPGEF3 required for the integration of small GTPase signaling into the MAPK cascade and the PI3K/Akt pathway, respectively. LPAR1 on the other hand is a receptor for the phospholipid lysophosphatidic acid (LPA) which can activate a variety of signaling pathways via G-proteins including MAPK and NF-κB, thereby regulating a variety of cellular functions like actin reorganization, migration, differentiation, and proliferation [47].

We decided to focus and experimentally tested the bioinformatically suggested stimulation of the PI3K/Akt and the MAPK pathways by investigating the levels of central effector proteins (Figure 4 and Figure 5). In the case of PI3K/Akt signaling, the PIP_3_ phosphatase PTEN is hyper-phosphorylated at Ser380 in the HeLa vtRNA1-1 knock-out cells at basal conditions and when starved, this effect is reversed upon re-feeding (Figure 4c). Ser380 phosphorylation attenuates PTEN activity and Akt is de-repressed. Inhibitory hyperphosphorylation of PTEN Ser380 was previously described in gastric cancer tissue [48] and hematopoietic tumors [49,50], representing a mechanism to enhance pro-survival PI3K/Akt signaling. Important cellular functions of c-Raf include downstream activation of MAPK signaling and mitochondrial localization and replacing Bad from the pro-apoptotic Bad/Bcl-2 complex [51,52]. The inhibitory c-Raf phosphorylation at Ser259 presents a binding site for the cytosolic adaptor 14-3-3 proteins, preventing c-Raf membrane accumulation and function [53]. The slight inhibitory hyper-phosphorylation on c-Raf S259 in the vtRNA1-1 knock-out cells at basal level (Figure 5c) might represent a prerequisite for the weak apoptotic resistance of these cells and a predisposition for elevated cell death. Through Ser338 phosphorylation of c-Raf the MAPK pathway is activated and c-Raf shows enhanced mitochondrial localization [52,54]. Our data from the ERK1/2 pathway analysis demonstrates enhanced c-Raf Ser338 phosphorylation when vtRNA1-1 expression is lost, which does not change upon short starvation (Figure 5c), but increases after 24 h of treatment (Figure 5b). This finding indicates that the loss of vtRNA1-1 is sufficient to induce the pro-survival Ser338 phosphorylation under normal growth conditions and under serum starvation. In addition, when cells are recovered from extended starvation treatment, only the vtRNA1-1 knock-out cells maintain the Ser338 phosphorylation of c-Raf longer (Figure 5b), thus hinting towards a greater need of pro-survival signals. The enhanced phospho-ERK1/2 levels in the vtRNA1-1 knock-out cells following serum starvation (Figure 5c) are indicative for a better cell survival of these cells. However, this is not what we observe (Figure 1e; Appendix A). It appears that vtRNA1-1 knock-out cells are already intrinsically challenged under optimal media conditions and thus upregulate basal levels of pro-survival programs. Upon nutrient deprivation these pathways get further activated in an attempt to cope with this increased, and putatively fatal, stress situation. In the absence of vtRNA1-1, however, cells fail to establish a robust signaling state required for establishing productive cellular homeostasis leading to, in the final consequence, an increased cell death rate. Apparently, more data is required to underpin this hypothesis and future work has to address the question how HeLa cells devoid of vtRNA1-1, despite the activation of several pro-survival counter measures, commit to apoptosis nevertheless when starved.

In an attempt to identify functional small vtRNA-RNPs, we affinity purified native in vivo assembled RNP particles using RAT-tagged [55] versions of vtRNA1-1 and vtRNA1-2 (Appendix A
Appendix A). Following depletion of cell extracts from the endogenous large 13 MDa vault complex, the tagged vtRNPs were affinity purified and associated proteins were identified by LC-MS/MS. Gene ontology (GO) analysis of the MS data (Appendix A) revealed proteins pulled-down preferentially with vtRNA1-1 that are involved in the regulation of GTPase activity and in the regulation of small GTPase-mediated signal transduction, functions fundamental to cellular signaling. Additionally, proteins negatively regulating the epidermal growth factor (EGF) receptor signaling pathway were purified rather with vtRNA1-1 than with vtRNA1-2. The EGF receptor belongs to the ErbB receptors that can regulate both the PI3K/Akt and the ERK pathways. Therefore, these MS findings are compatible with the mRNA-Seq analysis highlighting a role of cellular vtRNA1-1 levels in orchestrating pro-survival signaling pathways.

## 5. Conclusions

Cumulatively, our data confirms an essential and specific role of the vtRNA1-1 in apoptosis resistance in HeLa as well as in other immortalized human cell lines (A549, HEK293, BL2, BL41, HS578T). Genomic removal of the vtRNA1-1 gene resulted in modulation of signaling pathways known to regulate cell proliferation and cell death. Even though a direct physical link between vtRNA1-1 and members of these signaling pathways awaits future characterization, our study substantiates a pro-survival role of this ncRNA in human cancer cells.

## Figures and Tables

**Figure 1 biomolecules-10-00614-f001:**
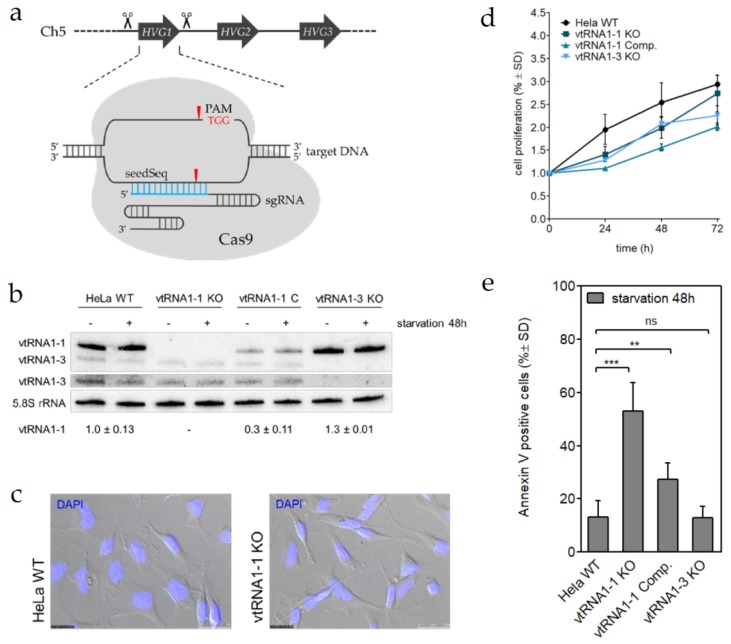
Vault RNA1-1 (vtRNA1-1) knock-out stimulates starvation-induced apoptosis. (**a**) Schematic representation of the genome editing strategy to remove a non-coding RNA gene. Up- and downstream cleavage sites are indicated. (**b**) Northern blot analysis was used to confirm successful genomic editing and vtRNA1-1 complementation in HeLa cell replicas. vtRNA levels were probed in cells either grown in complete media (−) or starved for 48 h (+). The vtRNA1-3 panel is shown a second time with enhanced contrast settings. 5.8S rRNA serves as internal loading control. (**c**) Light microscopic assessment of cell morphology. The bright-field picture was overlaid with the blue channel to visualize DAPI stained cell nuclei. (**d**) Growth curve over 72 h, measured with the MTT proliferation assay of HeLa WT, vtRNA1-1, and vtRNA1-3 knock-out cells and vtRNA1-1 complementation cells. MTT assay data shows the normalized mean values and the standard deviation from triplicates. (**e**) HeLa cells with (WT, vtRNA1-1 Comp.) or without (vtRNA1-1 KO, vtRNA1-3 KO) vtRNA expression were cultured for 48 h in starvation medium containing chloroquine and the amount of apoptotic cells was determined by FACS analysis after Annexin V and 7-AAD staining. This data is the mean and standard deviation of at least three independent experiments. The number of apoptotic cells in untreated control samples was subtracted from the starved samples and was usually <10%. Significant differences were determined using the two-tailed unpaired Student’s *t*-test (*** *p* < 0.0001, ** *p* < 0.001).

**Figure 2 biomolecules-10-00614-f002:**
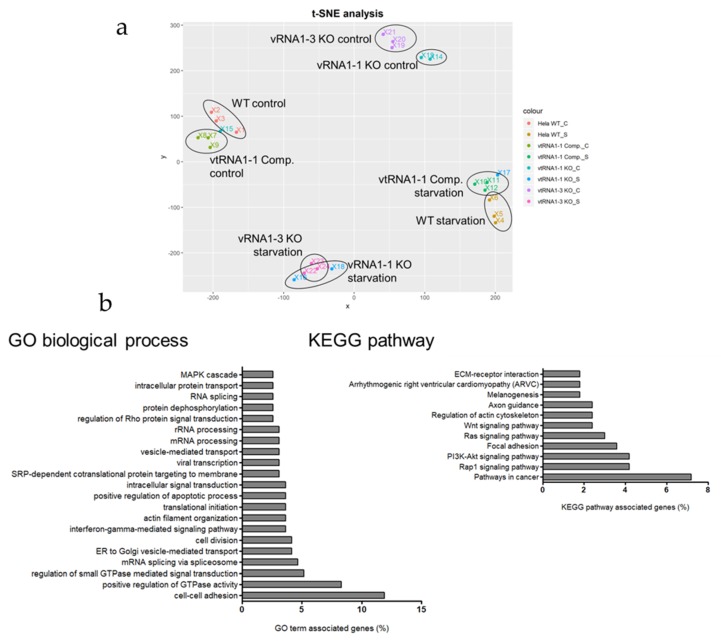
vtRNA1-1 loss leads to increased cellular signaling activity under starvation. (**a**) T-distributed Stochastic Neighbor Embedding (t-SNE) analysis of the mRNAseq data performed with all genes. vtRNA1-1 knock-out control sample #15 and vtRNA1-1 knock-out starvation sample #17 were excluded for all further downstream analyses. (**b**) Database for annotation, visualization, and integrated discovery (DAVID) gene ontology (GO) analysis and KEGG pathway analysis performed on 167 genes differentially expressed specifically in the vtRNA1-1 knock-out cells following 24 h of starvation with chloroquine. The functional annotation clustering application was used to determine important GO and KEGG terms.

**Figure 3 biomolecules-10-00614-f003:**
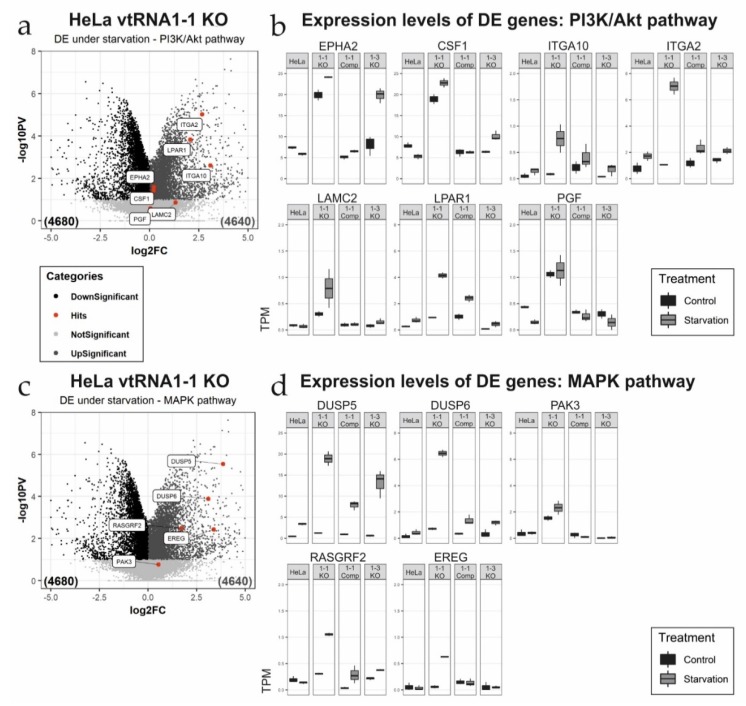
Differential expression analysis of genes involved in the PI3K/Akt pathway and the MAPK cascade. (**a**,**c**) Volcano plots illustrating differentially expressed (DE) genes comparing the vtRNA1-1 knock-out cells after 24 h starvation with chloroquine to the corresponding untreated control samples. Genes annotated either with the PI3K/Akt pathway (**a**) or the MAPK pathway (**c**), are highlighted (red dots) and flagged. (**b**,**d**) Box plots depicting the individual expression levels of the same genes highlighted in the volcano plots in all four cell lines used under control and starvation conditions. Gene transcript abundancies are indicated as transcripts per kilobase million (TPM) for the PI3K/Akt pathway (**b**) and the MAPK pathway (**d**), respectively.

**Figure 4 biomolecules-10-00614-f004:**
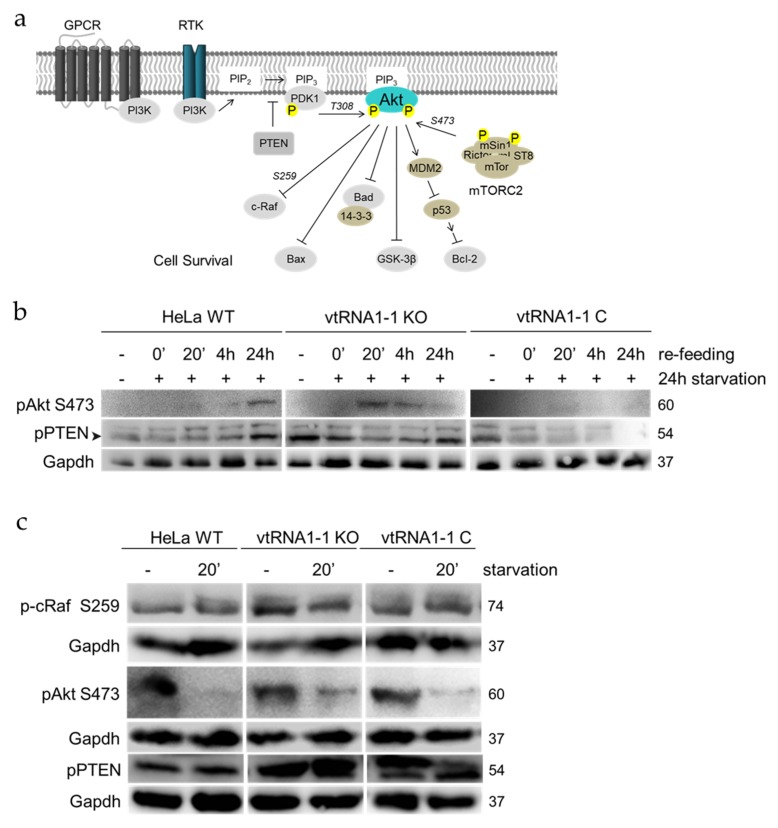
The loss of vtRNA1-1 stimulates PI3K/Akt signaling. (**a**) Simplified schematic representation of the PI3K/Akt signaling pathway. The kinase Akt is the main signal transducer. Important serine/threonine phosphorylations are indicated. (**b**) Western blot analysis was used to track the levels and the kinetics of mTORC2 phosphorylated Akt and phosphorylated PTEN as a consequence of 24 h starvation followed by re-feeding. Gapdh served as loading control. (**c**) Phospho-protein levels of the PI3K/Akt signaling involved proteins Akt and PTEN were assessed in the absence of starvation (−) and after 20 min of starvation with chloroquine (20′). Gapdh served as loading control.

**Figure 5 biomolecules-10-00614-f005:**
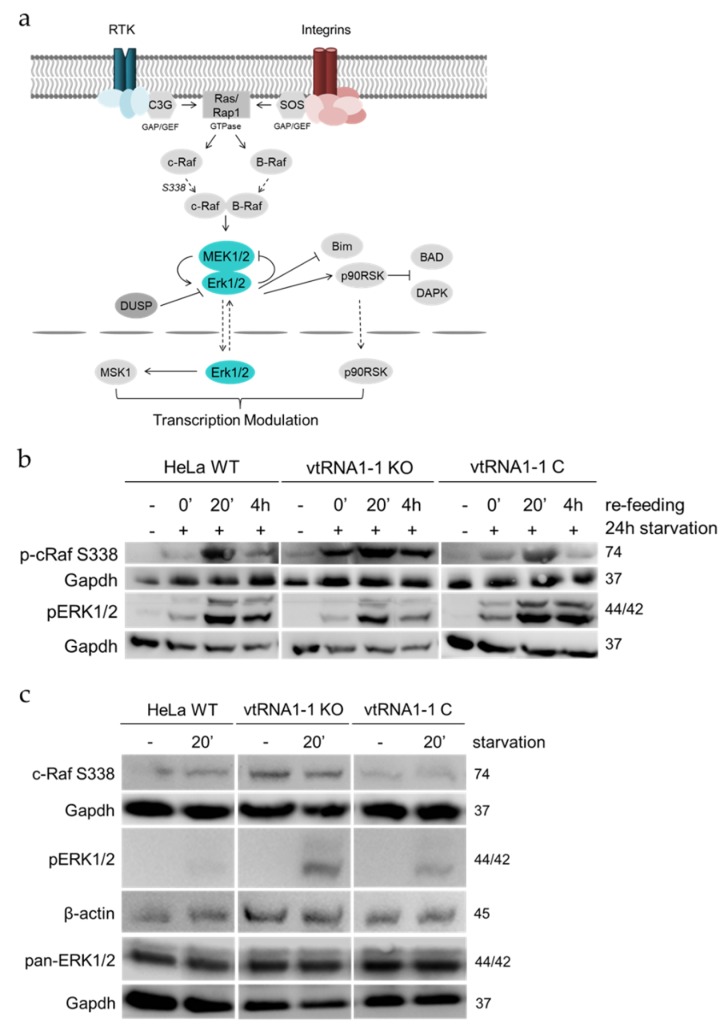
The initiation of the ERK1/2 MAPK cascade in starvation-induced HeLa cells is vtRNA1-1 dependent. (**a**) Simplified schematic representation of the ERK1/2 MAPK signaling cascade. The two kinases ERK1 and ERK2 are the major signal transducers. MEK1/2 represent the MAPKK and c-Raf represents the MAPKKK. (**b**) Western blot analysis was used to assay the levels and the kinetics of PAK1 phosphorylated c-Raf (S338) and phosphorylated ERK1/2 as a result of 24 h serum deprivation with chloroquine followed by re-feeding. Gapdh served as loading control. (**c**) Phospho-protein levels of the ERK1/2 cascade involved proteins c-Raf and ERK1/2 were assessed in the absence of starvation (−) or following 20 min of starvation with chloroquine (20′). Gapdh and β-actin served as loading controls.

**Figure 6 biomolecules-10-00614-f006:**
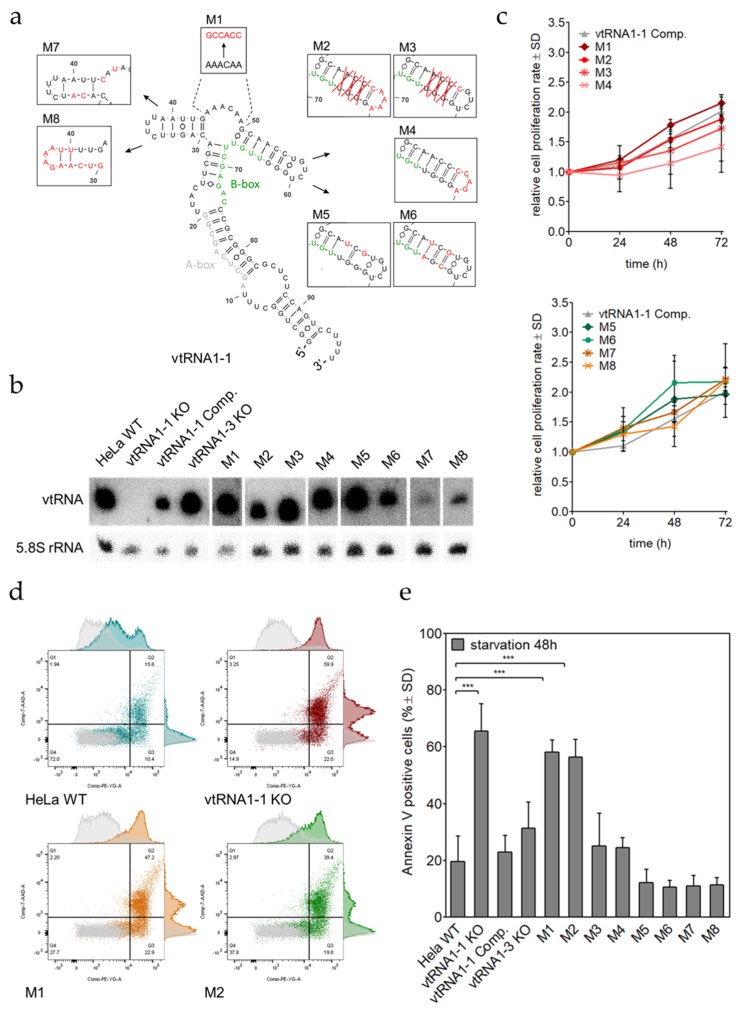
Twenty-four nucleotides within the vtRNA1-1 central domain are sufficient for apoptosis resistance. (**a**) A secondary structure model of vtRNA1-1 together with the schematic depiction of eight different mutations that were individually generated and by lentiviral transduction stably expressed in the endogenous vtRNA1-1 negative background of the vtRNA1-1 knock-out cell line. (**b**) The expression levels of M1 to M8 in HeLa vtRNA1-1 knock-out cells were confirmed by northern blot analysis with an oligonucleotide that was able to detect all vtRNA variants except M1, M7, and M8. For these mutants, individual DNA probes were designed. The 5.8S rRNA served as internal loading control. (**c**) Growth curve over 72 h, measured with the MTT proliferation assay of all mutant HeLa cell line derivatives. The vtRNA1-1 complementation cell line served as growth control. MTT assay data shows the normalized mean values and the standard deviation from triplicates. (**d**) After 48 h of starvation with chloroquine, HeLa cells expressing the mutant vtRNAs were stained with Annexin V and 7-AAD and apoptotic cells were quantified by FACS. The scatter plots depict HeLa WT, vtRNA1-1 knock-out, vtRNA M1, and vtRNA M2 cells without treatment (grey) and after 48 h of starvation with chloroquine (color). Quadrant 4 (Q4) represents unstained, viable cells, quadrant 3 (Q3) contains Annexin V positive, early apoptotic cells and quadrant 2 (Q2) gates Annexin V/7-AAD double-stained late apoptotic cells. (**e**) Quantification of Annexin V positive cell populations of the HeLa cells expressing vtRNA1-1 mutants after 48 h of starvation with chloroquine. HeLa WT, vtRNA1-1 knock-out, vtRNA1-1 complementation, and vtRNA1-3 knock-out cells served as apoptosis controls. The mean values and standard deviations of three independent experiments are shown. The Annexin V positive cell fractions of the untreated control cells were used to normalize the starvation samples and was usually <10%. Significant differences were determined using the two-tailed unpaired Student’s *t*-test *** *p* < 0.0001).

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
