# Peer review of "Human vtRNA1-1 Levels Modulate Signaling Pathways and Regulate Apoptosis in Human Cancer Cells"

_biomolecules, 2020, doi:10.3390/biom10040614_

Round 1

Reviewer 1 Report

This is an interesting manuscript.  While previous work has already shown that vault RNA1-1 modulates apoptosis, the underlying mechanism was not determined.  The current manuscript addresses this issue and reports that vtRNA1-1 helps to protect cells from apoptosis and acts through the PI3K/Akt pathway and the ERK1/2 MAPK cascade. Further, while the authors previously identified a critical central domain, they here have further narrowed the region within vtRNA1-1 that mediates this effect.  Overall, while the data largely extend previous observations, the work does represent an important advance in the field and a resource for future studies.  I do have several relatively minor issues that could be addressed.

  1. The authors carry out genomewide RNA-seq on cells that do or do not express vtRNAs and under different experimental conditions.  The data analysis identifies the desired signaling pathways but also reveal other perhaps important gene expression changes that are not discussed at all.  I feel it would be helpful to add a bit more meaningful discussion and analysis of other vtRNA effects on cells, since there appears to be a wealth of information in the data generated.  
  2. The proteomic analysis of potentially functional binding proteins to vtRNA1-1 could be expanded somewhat, as data has been generated and presented in a supplement.  

Reviewer 2 Report

First of all, congratulations on such a comprehensive work on the role of vtRNA1-1 in prolonged starvation conditions. I appreciate that it has been a hard and painstaking labour. However, showing so many experiments requires including multiple experimental details and, in this work some of them which are important for the understanding and reproducibility of the experiments are missing.

Some paragraphs of the text are too long, it would be better to describe them more precisely. Besides, an outline of the work or a graphical abstract would help.

Authors should indicate how efficiency validation of gRNAs has been carried out, and they should include a scheme of the CRISPR construct with the two gRNAs, at least in the supplementary material. Instead, Table 1 and Figure 1 should be removed from the supplementary material, since they correspond to previous work.

Regarding the mRNA-Seq analysis, why have you studied mRNA and not Total RNA? It would be interesting to see the role of other non-coding elements, don't you think?

Either way, I miss knowing the total number of sequences performed (reads per sample), and the deviation between them. Additionally, a description of quality control, mapping, and other pre-annotation bioinformatics analyses together with a list of over-expressed and silenced genes should be included in the supplementary material.

The authors must clarify the position of the genes in the "Volcano plots", (Fig. 3), they consider genes falling in the significant and non-significant area of the plot, what is the reason behind this selection? why these genes?

The results shown in the Western blot analyzes (Figure 4) are difficult to interpret, so these results make it difficult to understand the discussion. You have chosen pAktS473, why not Thr308?

Reviewer 3 Report

This manuscript studies the essential and specific role of the vtRNA1-1 in apoptosis resistance in several human cell lines. Genomic removal of the vtRNA1-1 gene resulted in modulation of signaling pathways known to regulate cell proliferation and cell death. Furthermore, the study identified a short stretch of 24 nucleotides of the vtRNA1-1 central domain as being essential for successful maintenance of apoptosis resistance. It is an interesting finding,

Major points:

  1. Please discuss the direct target genes of vtRNA1-1 in the ERK1/2 MAPK pathway and PI3K/Akt pathway, including the bioinformatics results or the experiment results.
  2. The 623 DE genes between HeLa WT and vtRNA1-1 knock-out cells and 633 DE genes between HeLa WT and vtRNA1-3 knock-out cells should be provided.

Round 2

Reviewer 2 Report

Acceptable to publish. The current version of the paper was improved significantly that deserves to be published by the journal.